# Smart Cities and Big Data Analytics: A Data-Driven Decision-Making Use Case

**Ahmed M. Shahat Osman \*** and **Ahmed Elragal**

Department of Computer Science, Electrical and Space Engineering, Luleå University of Technology, 971 87 Luleå, Sweden; ahmed.elragal@ltu.se
\* Correspondence: ahmed.mohamed.el-shahat@ltu.se

**Abstract:** Interest in smart cities (SCs) and big data analytics (BDA) has increased in recent years, revealing the bond between the two fields. An SC is characterized as a complex system of systems involving various stakeholders, from planners to citizens. Within the context of SCs, BDA offers potential as a data-driven decision-making enabler. Although there are abundant articles in the literature addressing BDA as a decision-making enabler in SCs, mainstream research addressing BDA and SCs focuses on either the technical aspects or smartening specific SC domains. A small fraction of these articles addresses the proposition of developing domain-independent BDA frameworks. This paper aims to answer the following research question: how can BDA be used as a data-driven decision-making enabler in SCs? Answering this requires us to also address the traits of domain-independent BDA frameworks in the SC context and the practical considerations in implementing a BDA framework for SCs' decision-making. This paper's main contribution is providing influential design considerations for BDA frameworks based on empirical foundations. These foundations are concluded through a use case of applying a BDA framework in an SC's healthcare setting. The results reveal the ability of the BDA framework to support data-driven decision making in an SC.

**Keywords:** big data analytics; smart cities; data-driven decision making; use case; voice of patients

## 1. Introduction

Governments and municipalities undertake smart city (SC) projects to mitigate the challenging impacts of continuous urbanization developments and increasing population density in cities. In general, SC projects aim at offering citizens a better quality of life in economically and environmentally sustainable cities [1–4]. The concept of a smart city is relatively modern for urban development. Although the term is used widely in academic literature and urban development projects, there is no common consensus about the meaning of "smart" in the context of SCs. Intuitively, an SC consists of smart components. The opposite is not valid, as smart components do not constitute an SC, but maintaining the city's components' interrelationship is an essential trait for a city to be smart. From the information systems (IS) perspective, this feature means collecting, analyzing, and interchanging data and information between various city domains [1].

Academics and practitioners tend to model SCs from the perspective of the smart domains constituting the city. However, the interrelationship between the underlying city domains is a crucial concept to realize city smartness. The interrelationship between the inter-domain systems is also an intuitively essential consideration. This integrated view of an SC implies cross-domain sharing of information. From this perspective, SC is viewed as a whole body of integrated systems or a system of systems [5]. Whereas information and communication technology (ICT) and IS are the key enabling technologies in SCs [2,4,6], this integral and holistic view of SCs reflects its complexities of the corresponding ICT and IS implementations.

The direct consequence of the intensive diffusion of ICT in SC domains (e.g., internet of things [IoT], mobile applications, smart meters) is the generation of vast volumes

of a mixture of structured, semi-structured, and unstructured digital data via machines, organizations, and people. This type of data is known as big data (BD). The analysis of these accumulated data (BDA) to extract latent insights and unknown information encourages academics and practitioners to support decision making in SCs [7–9]. The connection between SCs and BDA has been covered extensively in the literature [6,9–11]. Although there is an abundance of academic articles addressing the use of BDA in SCs, most research efforts focus on either the technological aspects, such as IoT implementations, or reaching *smartness* in specific SC domains (e.g., energy [12], environment [13], transportation [14]). Only a small fraction of these articles addresses the proposition of domain-independent BDA frameworks that can support a broad range of analytics in a multi-stakeholder environment, such as SCs [15–17]. These analytics frameworks do not also address how the interchange of information between various stakeholders in different SC domains and inter-domain stakeholders can be enabled. Taking into consideration the level of details and complexity of SC projects (either establishing a new city from scratch, such as Masdar in the UAE, or changing a legacy city into an SC, such as in Barcelona, Spain) in addition to the diversity of the stakeholders involved in these projects (from highly strategic decision makers to citizens and visitors), we can comprehend the need for resilient and efficient BDA mechanisms that can serve the requirements of various decision makers in SCs and at the same time allow the interchange of extracted analytics. The lack of research that addresses these concepts in data analysis systems is the motivation behind this article.

Through this paper, we attempt to fill this gap in the literature by answering the following research question: how can BDA be used as a data-driven decision-making enabler in SCs and at the same time allow the interchange of extracted analytics? To answer this, we conducted a systematic literature review of recent articles that address this research's main subjects: SCs and BDA. Twenty-six articles addressing BDA frameworks and applications in SCs are analyzed. Additionally, six BDA frameworks are discussed and compared. A prototype of one of the proposed BDA frameworks, the Smart City Data Analytics Panel (SCDAP) [10], is instantiated and applied in smart health use cases to analyze patient reviews and thus understand patients' concerns and satisfaction points. SCDAP is a domain-independent BDA framework. The SCDAP design supports the persistence of extracted analytics for later references by other stakeholders. In other words, it supports the interchangeability of extracted models between stakeholders.

This article has 10 sections. After this introductory section, the rest of the document is organized as follows. Section 2 introduces background information about the main subjects of this research: SCs and BDA, data-driven decision making, and decision-making levels. Section 3 presents the research method. A literature review is conducted in Section 4, including a discussion of the analyzed BDA frameworks. The importance of this research is presented in Section 5. The details of the BDA framework selected for prototyping (SCDAP) and its physical realization components are presented in Section 6. In Section 7, the steps in developing the evaluation use case, Voice of Patients (VoP), are presented. Sections 8 and 9 are the discussion of findings and the conclusion, respectively. Finally, the limitations of this research and extensions for future research are summarized in Section 10.

## 2. Background Information

In this section, we introduce the fundamental pillar subjects of this article: SCs, BDA, and decision-making. Uncovering the relationship between these subjects helps to highlight the importance of this type of research.

### 2.1. Smart Cities and Big Data Analytics

An SC is a well-performing city composed of smart domains. The pioneering work of Rudolf Giffinger defined six smart domains that characterize the smartness of an SC [3]. These are smart people, smart living, smart economy, smart governance, smart mobility, and smart environment, as depicted in Figure 1. Although academics have different views on the number and nature of smart domains that make up an SC, a common consensus is

that a city's smartnesss is realized by maintaining the cross-domain interrelationship and sharing of information [5]. Intuitively, this is achievable utilizing ICT-based solutions.

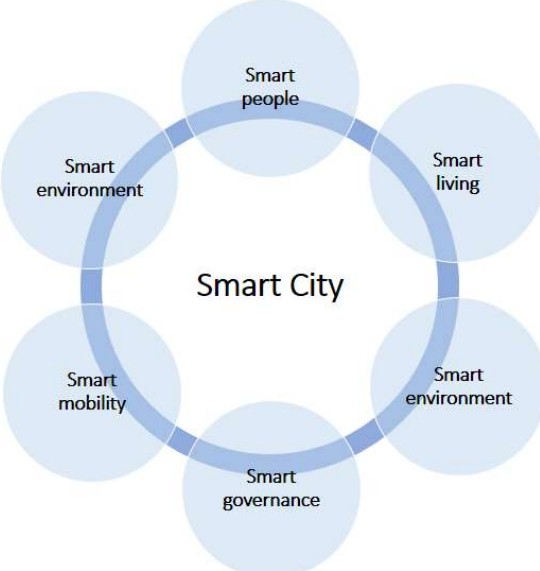

**Figure 1.** Smart city domains.

In return, the diffusion of ICT in various SC domains generates a wealth of vast data volumes. These data, known as BD, are a mixture of structured, semi-structured, and unstructured data [9,18]. The analysis of BD, called BDA, to extract insights and hidden correlations has received increasing attention in the field of IS to support decision-making [19]. The relevance of SC and BDA application has also drawn interest in the literature from different perspectives. Given the complex and dynamic nature of SC projects and the diversity of the involved stakeholders from the early stages of planning to the last stages of daily service monitoring and management, we can recognize the volumes of the accumulated BD during these stages in addition to the diversity of stakeholder requirements. Although there is a plethora of articles addressing the applications of BDA in different SC domains [20–26], few articles have presented propositions to develop domain-independent BDA frameworks that can serve the analytical requirements of stakeholders during different SC project stages [10,15,18] and at the same time preserve and interchange the extracted analytics.

## 2.2. Big Data Analytics and Data-Driven Decision-Making

BDA is a key enabler for data-driven decision making (DDD). DDD is a modern trend gaining increasing interest in IS; it is known as data science in its broad sense. DDD refers to the practice of basing decisions on the analysis of data rather than purely on intuition. The authors in [27] demonstrated the benefits of DDD and its positive impact on organizations' productivity. SCs are not an exception to this, as cities' smartness depends on the IS providing a wealth of diverse digital data that serve as a basis for decision-making through analytics [28,29]. Several algorithms are used in BDA, including artificial intelligence, machine learning, data mining, and deep learning [30]. Compared with the traditional statistical methods of data analysis, these modern methods are characterized by dealing with structured and unstructured data and extracting quantitative and qualitative indicators. Text analytics and image recognition are typical examples of unstructured data analysis.

Despite the abundance of scholarly articles that use BDA in SCs, the more significant part of research endeavors are centered on specific and explicit SC areas, such as energy [12], climate [13], or transportation [14]. Only a few of these articles address the recommendation

of developing end-to-end and domain-independent BDA frameworks that can uphold a broad scope of multi-stakeholder environments [10,15–17].

### 2.3. Decision-Making Levels

In an SC project, similar to a typical project, decisions are taken at three levels: strategic, tactical, and operational [31].

Strategic decisions are major choices of actions, and they influence the whole or major parts of a project or business enterprise. Such decisions are taken at high management levels. Strategic decisions contribute directly to the achievement of the common goals of the organization. They have long-term implications for the business enterprise. Traditionally, strategic decisions are unstructured, so a manager has to apply their business judgment, evaluation, and intuition to the definition of the problem. In this case, decisions are based on cumulative partial experiences and knowledge of environmental factors, which are uncertain and dynamic.

Tactical decisions are taken at the middle level of management. These decisions relate to the implementation of strategic decisions. They are directed toward developing divisional plans, structuring workflows, establishing distribution channels, and acquiring resources, such as manpower, materials, financials, and other resources.

Operational decisions are taken at lower levels of management. They are related to the day-to-day operations of the enterprise. They have a short-term horizon, as they are made repetitively. These decisions are based on facts regarding events and do not require much business judgment. As information is needed to help managers make rational, well-informed decisions, IS need to focus on managerial decision making.

## 3. Research Method

In this study, we applied two approaches. First, we conducted a systematic literature review of BDA frameworks within the context of SCs. The list of candidate articles are then analyzed according to the addressed SC domains. This step aims to survey the recently published articles proposing BDA frameworks in the context of SCs. This survey helps to analyze the research directions pertained to these two fields jointly. At the same time, it enables identifying the common traits of domain-independent BDA frameworks and comparing between these frameworks in the SC context. This step ends with selecting a framework that is domain independent and supports decision making with information exchange ability.

In the second step, a prototype for the selected BDA framework is instantiated using industry known software packages for empirical evaluation. To preserve the balance between technological and organizational dominance in designing and evaluating the framework prototype, we abide by the action design research (ADR) approach proposed in [32]. ADR consists of four stages, each anchored to a set of seven principles, as shown in Figure 2.

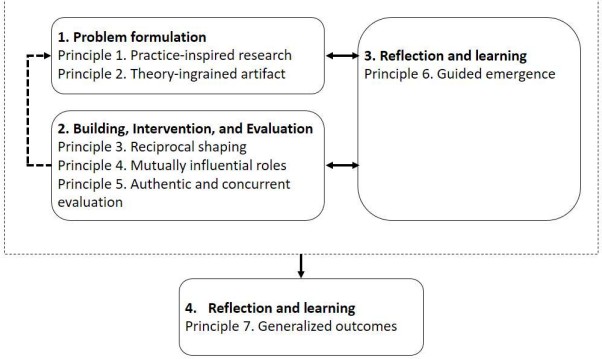

**Figure 2.** Action design research stages and principles [32].

- **Stage 1**: Problem Formulation

    This is the initial stage for a problem perceived in practice or anticipated by researchers where the initial research questions are set. The research opportunity is conceptualized based on existing theories and technologies. This stage draws on two distinct principles: (1) practice-inspired research and (2) theory-ingrained artifact.

    This stage is reflected in this article in both the introduction and background information sections. In these sections, the relevance of the research question "how can BDA be used as a data-driven decision-making enabler in SCs?" is set from practical and theoretical perspectives. The research gap that answering this question will cover is emphasized through the literature review in Section 4. Defining the problem at hand as an instance of a broader class of problems is a critical issue in this stage.

- **Stage 2**: Building, Intervention, and Evaluation (BIE)

    The second stage of ADR uses the problem framing and theoretical premises adopted in stage 1 as a platform for generating the initial design of the IT artifact. The IT artifact is further shaped by organizational use and subsequent design cycles as an iterative process interweaves Building, Intervention, and Evaluation (BIE). A significant characteristic of this stage is clarifying the locus of the novelty that might come from the organizational intervention's artifact design. This difference reveals two key choices influencing the research design ADR team, either IT-dominant BIE or Organization-dominant BIE. This stage draws on three distinct principles: (3) reciprocal shaping, (4) mutually influential roles, and (5) authentic and concurrent evaluation.

    Activities of this stage are applied in Sections 6 and 7, respectively. A prototypical artifact is instantiated and applied in a typical citizen-centered use case known as voice of patients or VoP, where patient reviews are analyzed for decision support purposes. The artifact's design principles and the outcomes of the use case are evaluated with relevant stakeholders (i.e. organization-dominant BIE).

- **Stage 3**: Reflection and Learning

    This stage recognizes that the research process involves conscious reflection on the problem framing, the theories chosen, and the emerging ensemble artifact is critical to ensure that contributions to knowledge are identified. It is also essential to adjust the research process based on early evaluation results to reflect the ensemble artifact's increasing understanding. This stage is drawn on only one principle: (6) guided emergence.

    This stage is reflected in Section 8 where the design principles and generalizability of the artifact are discussed in the light of stakeholders' feedback analyses.

- **Stage 4**: Formalization of Learning

    The objective of the fourth stage of ADR is to formalize the learning. Casting the problem-instance into a class of problems (see Stage 1) facilitates the conceptual move into this problem's generalized class. Researchers outline the accomplishments realized in the IT artifact and describe the organizational outcomes to formalize the learning. These outcomes can be characterized as design principles (DP) and with further reflection, as refinements to theories that contributed to the initial design (principle 2). This stage draws on one principle: (7) generalized outcomes.

    This stage's principle is demonstrated in Sections 8 and 9, where the generalizability and concluded learnings of the artifact as a solution for a broad class of problems is discussed and demonstrated.

## 4. Literature Review

This section aims to answer the first research question addressing the traits of domain-independent BDA frameworks in the SC context. This literature review followed the [4] scheme proposed by [33] and adapted by [34]. This scheme includes six literature review characteristics: (1) focus, (2) goal, (3) organization, (4) perspective, (5) audience, and (6) coverage.

- Focus: Research outcomes and applications
- Goal: Integration and criticism
- Organization: Conceptual
- Perspective: Neutral
- Audience: Specialized scholars, practitioners, and SC planners
- Coverage: Exhaustive with selected citations (covers the period from 2018 to 2020)

### 4.1. Literature Review and Synthesis

Five IS online databases, ACM DL, IEEE, SCOPUS, Springer Link, and INSPEC, are searched using two search keywords, "big data" and "smart city," to collect quality scholarly articles. The search is limited to articles published in English for the period from 2018 to 2020. The number of research articles returned from each database is listed in the table below (Table 1):

**Table 1.** Search databases and number of hit articles.

| Database | No. of Articles |
| --- | --- |
| ACM DL | 6 |
| IEEE | 21 |
| SCOPUS | 35 |
| Springer Link | 19 |
| Science Direct | 28 |
| INSPEC | 19 |
| Total | 128 |

The total number of returned articles is 128. A filtration process is applied to eliminate duplicate articles from the analysis process, resulting in the exclusion of 34 articles. The rest of the 94 articles are further filtered by evaluating each article's title, abstract, and conclusion to assess how each article is relevant to the subject of this research. In some cases, a full article is reviewed if the prior steps are not sufficient, resulting in the removal of 68 more articles and the final 26 articles for analysis. The results are listed in Appendix A.

### 4.2. Analysis with Respect to Smart City Domains

The table presented in Appendix A shows domain-independent and domain-specific solutions. Note that only six articles (23% out of the 26 articles) address the subject of domain-independent BDA for SCs. The remaining 77% of the articles address specific SC domains. These ratios align with our prior claim that most research has focused on BDA in specific SC domains, whereas fewer studies have addressed domain-independent BDA frameworks. To address the current paper's objectives, we focus our analysis in this subsection on the domain-independent analytics frameworks introduced in these papers.

The authors in [10] introduced a domain-independent BDA framework called SCDAP. SCDAP is a three-layer domain-independent end-to-end BDA framework for SCs [10]. The logical design of SCDAP is based on six design principles: a layered design approach, standardized data acquisition and access, realtime and historical data analytics, iterative and sequential data processing, extracted model management, and aggregation.

The last two principles enable maintaining the models extracted from the data analysis. This feature has two benefits. First, creating a repository for the extracted models can be considered a form of knowledge memory for the SC. Second, this feature enables stakeholders and decision makers to share the analysis and knowledge results between them. This feature meets the goal of maintaining the cross-domain interrelationship of SC domains and, intuitively, inter-domain system interrelationship.

In [17], the authors presented a generic four-tier BDA framework comprising the sensing hub, the storage hub, the processing hub, and the application. The presented framework is a typical BDA framework that is similar to the frameworks proposed in other articles. Although the authors identified some key challenges in leveraging BDA in SCs,

they did not present how the proposed framework is different in handling these challenges. Furthermore, no real realization of the framework or use case was presented.

In [30], the authors presented a proposal for a six-layer BDA framework. The proposed framework's core data analysis engine utilizes two computational techniques: deep learning neural networks and fuzzy logic in data analytics. The utility of the proposed framework is demonstrated using the taxi demand prediction case study discussed. The classification results provided by the proposed framework contribute to optimizing the realtime distribution of taxis based on the predicted demand. This optimization directly affects other aspects, such as improving taxi availability, reducing waiting and journey times, and minimizing $CO_2$ emissions.

In [35], the authors introduced a general architecture based on IoT and BDA for disaster-resilient smart cities (DRSCs). The main feature of DRSC is providing disaster management with early warnings through collection, integration, and analysis of realtime and offline data from heterogeneous city data resources. Additionally, DRSC enables the prediction and monitoring of disaster situations. DRSC is implemented using a combination of Hadoop and Spark engines. It is evaluated in terms of processing time and throughput with simulated data generated from (a) a fire dynamic simulator, (b) gas sensors for pollution monitoring, (c) road traffic simulators, and (d) Twitter data (one-month crowdsourced data about disasters).

In [36], the authors proposed a domain-independent conceptual framework built on their previously proposed framework called Frame, Pixel, Place, and Event (FrAPPE). The idea behind FrAPPE is built on the digital image processing metaphor. Events are recorded in a time series of frames ($\tau_{n-1}$, $\tau_n$, $\tau_0$, $\tau_1$, $\tau_2$ ... $\tau_m$, $\tau_{m+1}$ ... ). Each frame includes information about the event location, in which the location is logically represented as hierarchical levels of grids and pixels. The main feature of FrAPPE is its ability to easily track and analyze a sequence of events in terms of time and location and then predict what potentially happens. The framework is implemented using the Hadoop ecosystem (e.g., HDFS, YARN, HIVE, P.I.G., Spark SQL, Spark Streaming, and SparkR). Its feasibility, generality, and effectiveness are demonstrated through multiple use cases and examples taken from real-world requirements collected in various cities (e.g., Milano Design Week, Milano Fashion Week, and Milan Expo 2015).

Lastly, the BDA Framework for Smart City (BDAFSC) proposed in [37] is a classical general-purpose BDA framework. The authors in [37] presented an idea about the processing of massive historical data in parallel to realtime data analysis, emphasizing data quality control. From the technological standpoint, BDAFSC is implementable using the Hadoop ecosystem: (a) data acquisition: KURA; (b) data transmission: Kafka, Flume, and Zookeeper; (c) preprocessing, batch processing: Spark; (d) data storage: HDFS HBASE; and (e) streaming component: Spark Streaming.

*4.3. Discussion*

By comparing the previously reviewed BDA frameworks in Section 4.2, we realize that these frameworks share some common characteristics, such as adopting the cascaded-layer design approach, enabling realtime and historical data analysis, and using horizontally scalable platforms, such as Hadoop. On the other hand, there are variations in some other features, such as dealing with data acquisition to cope with the diversity of input data and the nature of applied data analysis mechanisms. Moreover, SCDAP includes two special functionalities to manage extracted analytics, namely model management and model aggregation [10]. These two functions enable persisting the extracted analytics (i.e., data models) in a special repository for later reference. Although these two functionalities characterize SCDAP compared with other reviewed frameworks, SCDAP can be considered an extension of these frameworks. However, the authors in [10] did not indicate the actual implementation of real or synthetic SC use cases. The following table (Table 2) summarizes the characteristics of the reviewed BDA frameworks. This table answers the first research question on the features of domain-independent BDA frameworks in the context of SCs.

**Table 2.** Traits of the reviewed frameworks.

| | Trait/Framework | SCDAP [10] | [18] | [30] | DRSC [35] | FrAPPE [36] | BDAFSC [37] |
|---|---|---|---|---|---|---|---|
| 1 | Cascade layer design | Yes | Yes | Yes | Yes | Yes | Yes |
| 2 | Horizontally scalable platform (e.g., Hadoop) | Yes | Not defined | Not defined | Yes | Yes | Yes |
| 3 | Interoperability (data access/acquisition) | Yes | Not defined | Not defined | Yes | No | Yes |
| 4 | Realtime data analysis | Yes | Yes | Yes | Yes | Yes | Yes |
| 5 | Historical data batch analysis | Yes | Yes | Not defined | Yes | Yes | Yes |
| 6 | Extracted data model management | Yes | No | No | No | No | No |

## 5. Importance of the Research

This research is important both from an academic perspective and an empirical one. First, it represents a novel research direction toward the applications of BDA frameworks in SCs. Second, from a practical point of view, establishing a repository to persist the extracted analytics and data model resulting from the analysis process is a novel idea that characterizes SCDAP compared with other BDA frameworks. SCDAP is a domain-independent framework that can be applied in different SC domains; the model repository will contain the outcome of the various analytical methods and the feedback from various stakeholders. This repository will represent a knowledge repository or knowledge library where extracted analytics and models are systematically captured, organized, and categorized (Figure 3). In this sense, knowledge repositories help connect decision makers and other SC stakeholders via searchable libraries. This feature aligns with the preservation of the interrelationship of SC domains and inter-domain system interrelationship.

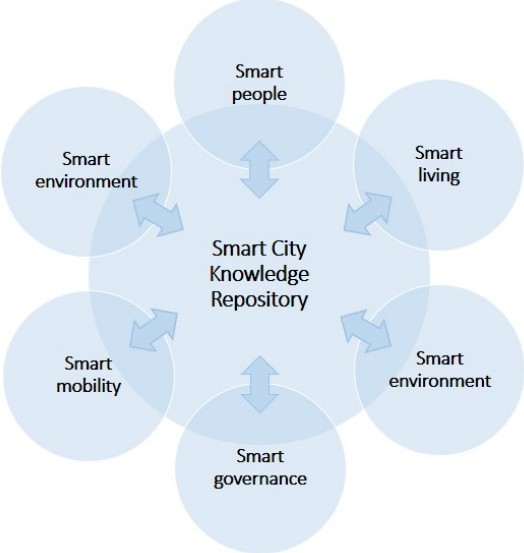

**Figure 3.** Smart city knowledge repository.

## 6. SCDAP—Prototypical Instantiation

In this section, we present the physical implementation of the SCDAP platform using Hadoop. Before delving into the details of the prototype instantiation of SCDAP, we provide an overview of SCDAP design principles. We will also review the comments and critiques mentioned in papers that cited SCDAP (88 articles). This step aims to identify any required modifications or flaws that need to be considered before the actual instantiation of SCDAP.

### 6.1. Smart City Data Analytics Panel (SCDAP)

SCDAP is a domain-independent end-to-end BDA framework for SCs [10]. The logical design of SCDAP is based on six design principles: a layered design approach, standardized data acquisition, access, real-time and historical data analytics, iterative and sequential data processing, extracted model management, and aggregation (Figure 4).

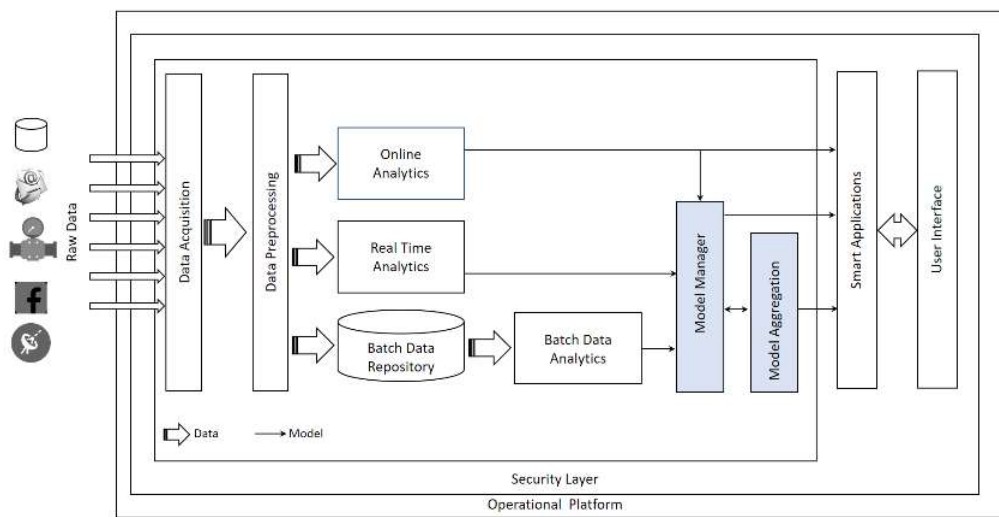

**Figure 4.** SCDAP conceptual design schematic diagram [10].

The two features that characterize SCDAP has two benefits. First, creating a repository for the extracted models can be considered a form of knowledge memory for the SC. Second, enabling stakeholders and decision makers to share the analysis and knowledge results between them. This feature applies to the interrelationship of SC domains and inter-domain system interrelationship, fulfilling the fundamental concept of smartness.

### 6.2. Commentaries on SCDAP

In the following bullets, we summarize the critical comments on the SCDAP framework, followed by an annotation on how we address each one in this research:

- The necessity of considering the urban platform's holistic view as an SC solution [38].
    - The SCDAP design is based on six design principles [10]. It is designed as a domain-independent BDA analytics framework to serve the holistic view of SCs. The main features of SCDAP compared with other frameworks or the proposed architectures are the introduction of model management and model aggregation functionalities to build a kind of model memory or library for the SC. This idea is emphasized in [11], where the authors mentioned the following clearly: "*Hence, the database should include a knowledge base module in which facts (observed) and rules for inferencing consequences and adaptation policies are stored separately from 'data' used for general computations.*" The concept of having a knowledge base for extracted facts is reflected in Figure 4 below (model repository). Figure 5 shows the updated conceptual design of the SCDAP three-layer architecture; it has (a) a platform layer, (b) a security layer, and (c) a data processing layer.

- Indications of the actual application and performance on real or synthetic SC data [37,39].
  - A prototype of SCDAP is instantiated and evaluated in this article.
- The addition of functionalities to monitor data quality during all phases of SCDAP [37].
  - Data quality is maintained in the Data Pre-processing module, where data selection, cleansing, transformation, and integration functionalities are executed. After completion of the analysis processes, the extracted information (in the form of data models) could be persisted for later reference according to their significance to end-users.
- The development of efficient algorithms for processing and extraction is not explained [37].
  - SCDAP is not meant for developing efficient algorithms; this is out of the SCDAP's scope.

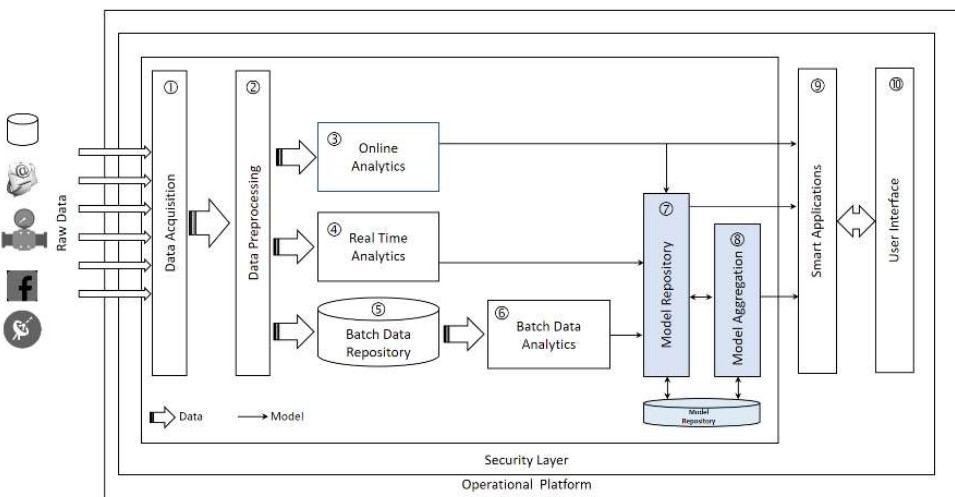

**Figure 5.** SCDAP conceptual design (updated).

*6.3. SCDAP Prototype Instantiation*

To instantiate an SCDAP prototype, we utilize the single-node Cloudera Distribution of Hadoop (CDH) QuickStart VM. CDH is a limited-version implementation of the full-suite Cloudera Hadoop. It is designed mainly for experimental and proof-of-concept purposes.

CDH includes the necessary components to construct the SCDAP platform layer. However, within this article's scope, we will focus only on the components that serve the design of both the platform layer and the data processing layer. The basic CDH QuickStart VM components that serve the design of the platform layer include the following:

- YARN (Yet, Another Resource Negotiator): YARN enables Hadoop to run interactive queries, realtime applications, and batch jobs simultaneously on one shared dataset. It isolates resource management and scheduling from the data processing components.
- MapReduce: A flexible parallel data processing framework for large datasets.
- HDFS (Hadoop Distributed File System): This is the main Hadoop data management system. It is a high-performance, scalable, distributed, fault-tolerant, and reliable data storage system. HDFS is designed to span large clusters of commodity servers and manage large volumes of data files. HDFS allows file creation, write once, read many, and remove operations. It does not allow update operations.
- HBase: This is a NoSQL column-oriented database in which every column (or family of columns) is treated individually.
- Hive: This is a high-level language built on top of MapReduce to analyze large datasets.

- Cloudera Impala: It provides fast and interactive SQL queries on data stored in HDFS or HBase. Impala uses the same storage platform, metadata, SQL syntax (Hive SQL), ODBC driver, and user interface as Hive.
- Apache Spark: Spark is a batch in-memory computing framework that can perform micro-batch procession via Spark Streaming. Spark offers fast performance given memory requirement considerations as a faster alternative to the Hadoop MapReduce programming framework.
- Apache Flume: Flume is an efficient distributed service for collecting, aggregating, and moving large volumes of log data for streaming into Hadoop. Flume's main use case is ingesting data into Hadoop.
- Apache Kafka: Kafka is a distributed publish-subscribe message streaming platform. It is used to build realtime streaming data pipelines that reliably transmit data between systems or applications (Hadoop is one of them). It also allows the processing of data stream records as they occur.
- Apache Sqoop: Sqoop is a tool designed to transfer bulk data between relational databases (e.g., MySQL, Oracle) into the HDFS and vice versa.

### 6.3.1. The Data Processing Layer

The data processing layer comprises the necessary components for data acquisition, preprocessing, analysis, and presentation. In this layer, we use RapidMiner (https://rapidminer.com/). RapidMiner is a data science software platform that provides a comprehensive and robust library of data analysis and built-in processing components, known as operators. In addition to RapidMiner's built-in library of operators, it extends its library with an abundance of third-party operators (e.g., text mining, JSON parsers, deep learning, Python scripting.) Although RapidMiner operators expedite the development cycle of data analysis processes through ready-made operators, RapidMiner is powered with Python and R scripting operators to add additional functionalities.

Radoop is one of the key RapidMiner operators' enabling seamless connectivity to Hadoop. Radoop extends RapidMiner's library with more than 60 operators for data transformations and advanced predictive modeling that runs on a Hadoop cluster.

Figure 6 shows the physical realization mapping of the SCDAP prototype using CDH and RapidMiner. The components of SCDAP physical realization in Figure 6 are labeled with numbers analogous to the corresponding conceptual design components in Figure 5. Lastly, the extracted results are presented to end users using Microsoft Power BI dashboards.

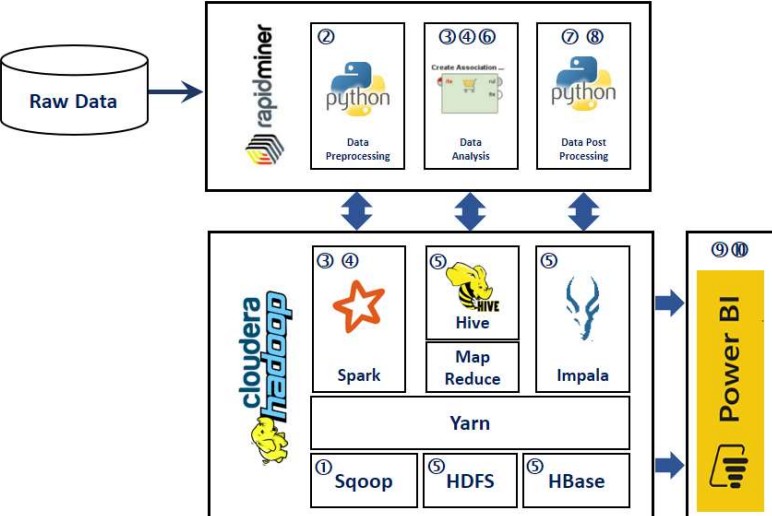

**Figure 6.** SCDAP prototype physical realization using CDH, RapidMiner, and Power BI.

### 6.3.2. Data Preprocessing

Data preprocessing functionalities are implemented using Python scripts running through the RapidMiner Python operator. These functionalities include tasks to remove anomalies from raw data in preparation for the analysis stage. Data preprocessing includes the following:

- Removing special characters from username, business name, and user reviews
- Data cleansing: Filling in missing values and smooth noisy data, identifying or removing outliers, and resolving inconsistencies
- Data integration: Integration of multiple databases, data cubes, or files
- Data reduction: Dimensionality reduction, numerosity reduction, and compression
- Data transformation and discretization

### 6.3.3. Data Analysis

RapidMiner data analysis operators are the core analysis components to ease and reduce development time. Within the scope of this article, we focus mainly on Meaning Cloud extension operators. Meaning Cloud is a market-leading software for text analytics and semantic processing. It empowers RapidMiner with an easy-to-use and powerful suite of text analytics operators (e.g., sentiment analysis, aspect-based sentiment analysis, topic extraction).

### 6.3.4. Data Post-processing and Model Management

The output data from the analysis phase might be generated in a form not suitable for end-user presentation and model management (e.g., not normalized, redundant). Additional functionalities are required to restructure these output data, if needed, before storage for the following stages.

Post-processing includes model management and aggregation functionalities that enable end-users to manage the extracted model's metadata through detailed data entry and retrieval screens.

## 7. SCDAP Use Case: Voice of Patients (VoP)

Improving citizens' quality of living is one of the main goals of adopting SC projects. The ability to monitor and evaluate citizens' satisfaction with the provided services that touch their daily lives enables decision makers to better understanditizens' satisfaction and hence make fact-based decisions [40–42]. Traditionally, collecting citizens' feedback is usually performed through semi-structured questionnaires and opinion polls. At present, social networks (e.g., Facebook, WhatsApp, WeChat) and Web 2.0 applications (e.g., www.smartservicedesk.com, www.fixmystreet.com, www.qlue.co.id, and www.yelp.com) play this role effectively by enabling citizens to express their opinions freely, mostly written as free text. Applying text analytics techniques to examine citizens' reviews is a modern trend in understanding citizens' opinions about the services provided to them [40,43]. This trend is often called Voice of Citizens (VoC). VoC can be viewed as a two-way street; on one hand, it is a worthy channel for evaluating citizens' experience for better decision making. On the other hand, it involves citizens in decision- making. These techniques have an increasing role in decision-making support [44,45]. A relevant application of this concept is the VoP. VoP refers to utilizing patients' opinions, experiences, and reviews to inform medical treatment.

As an evaluation use case for SCDAP, we apply VoP as a use case to demonstrate the feasibility of using SCDAP in analyzing patients' reviews and supporting decision-making with quantitative figures about patients' satisfaction with their services.

### 7.1. Approach and Data Sources

Text analytics is the main approach to analyze citizens' reviews and then convert the resulting figures into quantitative indicators. Text analytics has many objectives, but for the VoP use case, we focus on sentiment analysis and aspect-based sentiment analysis only.

Sentiment analysis allows identifying citizens' sentiments (positive, negative, or neutral) toward services, products, or brands [43,46]. Aspect-based sentiment analysis provides fine-grained sentiment information about various aspects composing the text [47].

In the VoP use case, we use citizens' reviews recorded through www.yelp.com as the main data source. The Yelp website and the Yelp mobile application enable crowdsourced reviews about businesses and services. For the sake of educational and academic research purposes, Yelp provides sample datasets in JSON format. These sample datasets include 8 million reviews from 2 million users on about 200,000 businesses. The operational entity-relationship diagram (ERD) model of the sample dataset is shown in Figure 7.

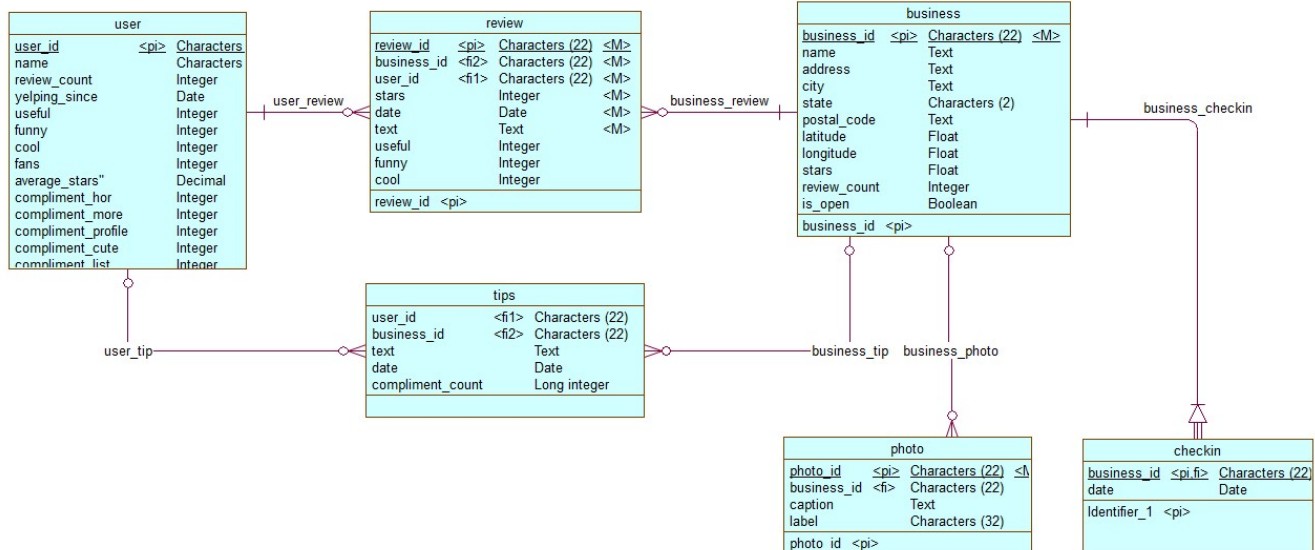

**Figure 7.** ERD model of the Yelp sample dataset.

In the following two subsections, data preprocessing and data analysis, the focus will be on the user, business, and review datasets as the three main datasets for analysis.

*7.2. Data Preprocessing*

In the first step of data preprocessing, raw data from Yelp (in JSON format) are loaded into the MySQL database with the relational schema shown in Figure 7. Preprocessing tasks are applied to the three main datasets: user, business, and review. It is worth mentioning that not all five data preprocessing tasks are applied to the three datasets. In general, special characters are removed from all text attributes (e.g., username, user review). The integrity between the three datasets is checked to exclude orphans and widow records.

user—includes the following basic attributes about reviewers:

| | |
|---|---|
| user_id | : Unique user id (22 character) |
| name | : Username |
| review_count | : Number of reviews written by the user (reflects how active the user is) |
| yelping_since | : String formatted as YYYY-MM-DD, indicating when the user joined Yelp |

Although the user entity includes detailed counts indicating the level of user interaction with other reviews, it does not include the user's gender, which is an important sentiment classifier. To predict the user's gender, the user dataset is integrated with a third-party service (https://genderize.io) to predict the user's gender from their name. The user's gender is predicted through (https://genderize.io) API. This service returns the predicted gender with two figures; the probability indicates the certainty of the assigned gender and the count of data rows examined to calculate the response.

business—includes the following basic attributes about the business:

| | |
|---|---|
| business_id | : Unique string business id (22 characters) |
| name | : Business name |
| address | : Full address of the business |
| city | : City |
| state | : Two-character state code (for US states) |
| postal code | : Postal code |
| latitude | : Latitude |
| longitude | : Longitude |
| stars | : Star rating, rounded to half-stars |
| review_count | : Number of reviews |
| is_open | : 0 or 1 for closed or open, respectively |
| attributes | : Business attributes to values (e.g., garage, parking) |
| categories | : Business category classification (e.g., restaurant, clinic, club) |
| hours | : Opening hours |

Entries of business categories are compiled in a normalized entity (category) to exclude redundant entries and are classified into one of the following services:

1. Business
2. Communication
3. Construction and Engineering
4. Distribution
5. Education
6. Environment
7. Finance
8. Tourism
9. Health
10. Recreation
11. Transportation
12. Other

The many-to-many relation between (business) and (category) is reflected in a joint entity: (business-category).

review—contains full-review text data, including the user_id (FK) of the person who wrote the review and the business_id (FK) the review is written for. This dataset includes the following attributes:

| | |
|---|---|
| review_id | : 22-character unique review id |
| user_id | : 22-character unique user id (user FK) |
| business_id | : 22-character business id (business FK) |
| Stars | : Star rating |
| Date | : Review date formatted (yyyy-mm-dd) |
| text | : The review text itself |
| useful | : Number of useful votes received |
| Funny | : Number of funny votes received |
| Cool | : Number of cool votes received |

The only preprocessing applied to the review dataset is the removal of special characters from the review text itself. The final ERD model used for analysis is shown in Figure 8. Note: tips, photos, and checkin entities are discarded from the analysis.

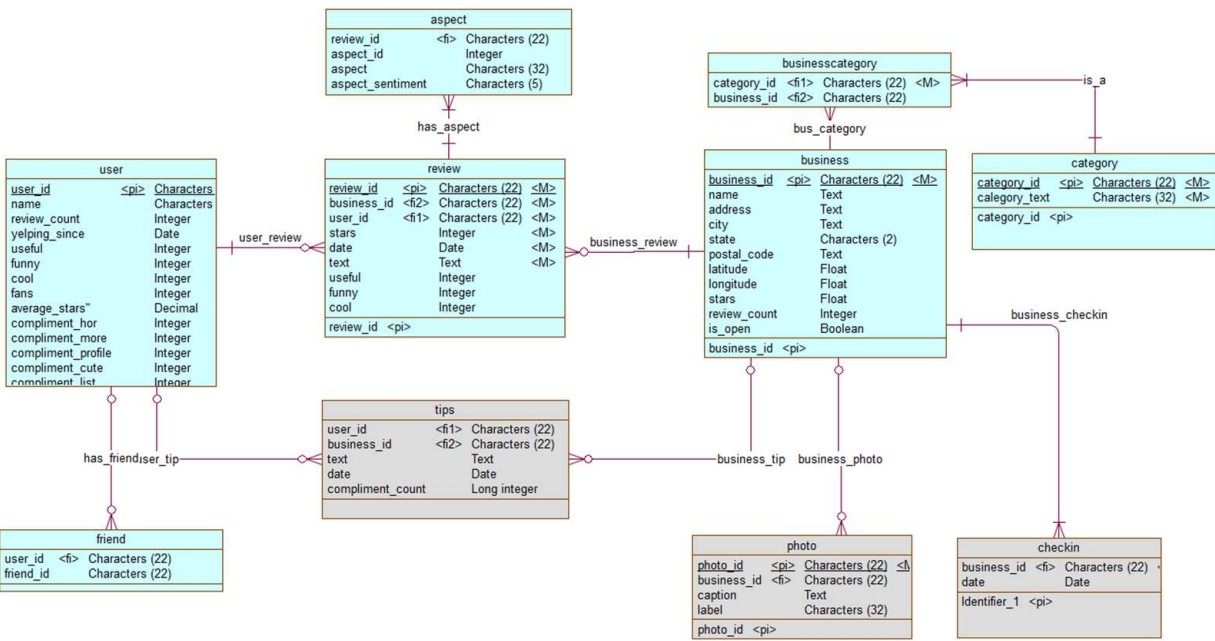

**Figure 8.** ERD model for analysis.

## 7.3. Data Analysis

**Step 1**: Review's sentiment analysis

The review text field's sentiment analysis is performed using MeaningCloud's Sentiment Analysis operator for RapidMiner. This operator returns six sentiment levels:

- P+: strong positive
- P: positive
- NEU: neutral
- N: negative
- N+: strong negative
- NONE: without sentiment

This operator returns the following supplementary detailed information about the analyzed text, such as subjectivity/objectivity and confidence percentage.

**Step 2**: Review's aspect-based sentiment extraction

To extract more granular sentiment reviews, we use MeaningCloud's Aspect-based Sentiment Analysis operator. This operator returns the sentiment of each aspect with customer reviews.

**Step 3**: Output data post-processing

The output data from MeaningCloud's aspect-based sentiment extraction step comes in a non-normalized form as long string concatenating each extracted aspect with its associated sentiment value. This string is processed (tokenized at the aspect level) to obtain a normalized form (reflected as an aspect entity in Figure 7).

**Step 4**: Presentation of results

To visualize and present the results, we used Microsoft Power BI (free edition). All figures presented in the following section are screenshots of the dashboards developed by Power BI.

The complete realization of SCDAP for VoP implementation using the above software components is shown in Figure 9.

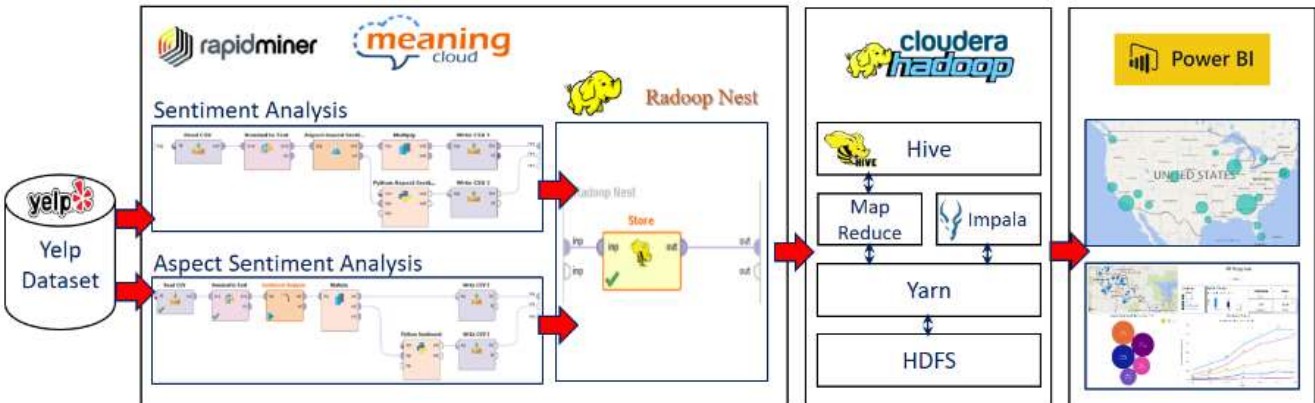

**Figure 9.** SCDAP for VoP Physical Realization.

**Step 5**: Model management and aggregation

After the analysis processes are completed and satisfactory results are extracted, metadata about the extracted resultant data models are persisted in a special database. Persisted results (referred to as resultant models) can be retrieved with relevant metadata for future inquiries or are aggregated for higher analytics levels. A JSON-like message for the meta-data describing the extracted resultant data models is shown below:

```
{
"model": {
    "model_id": "VoP-0001",
    "model_name": "Yelp-VoP-M0001",
    "project_name": "Smart City Big Data Analytics Panel SCDAP"
    "city_name": "usa-canada",
    "creation_date": "2020-12-01",
    "last_update": "2020-12-30",
    "domain": "healthcare",
    "authors": ["author 1", "author 2", "author 3", "author 4", "author 5", "author 6"],
    "keywords": ["kword 1", "kword 2", "kword 3", "kword 4", "kword 5", "kword 6"],
    "source_dataset": "yelp_json",
    "analysis_server": ["rapidminer", "meaning_cloud"],
    "model_folder": "extracted model folder path",
    "model_workbook": "vop-workbook",
    "model_description": "this is a long text description for the extracted model"
    }
}
```

*7.4. Analysis of Patient Reviews*

Figure 10 shows the main VoP dashboard. This dashboard enables decision-makers to select a hospital (or a group of hospitals) in order to track patient reviews' changes in sentiment levels over the past years (lower right corner). The top five aspects are displayed on the bubble chart in the lower-left corner. For a more detailed analysis, the user can select specific levels of sentiments from the sentiment slicer.

For the example shown in Figure 11, the selected hospital has received a three-star average evaluation over the past seven years. However, the sentiment graph shows a significant decrease in the positive reviews during the last three years (25 ▶ 16 ▶ 12) (i.e., a decrease by ~50%). In the last two years of the same period, there was a remarkable increase in the negative sentiments (10 ▶ 25) (i.e., an increase by ~50%). These figures reflect the deterioration of patient satisfaction in the last three years. To obtain more detailed insights into aspects of patients' interest, negative, and very negative sentiments are marked on the sentiment slicer (Figure 12).

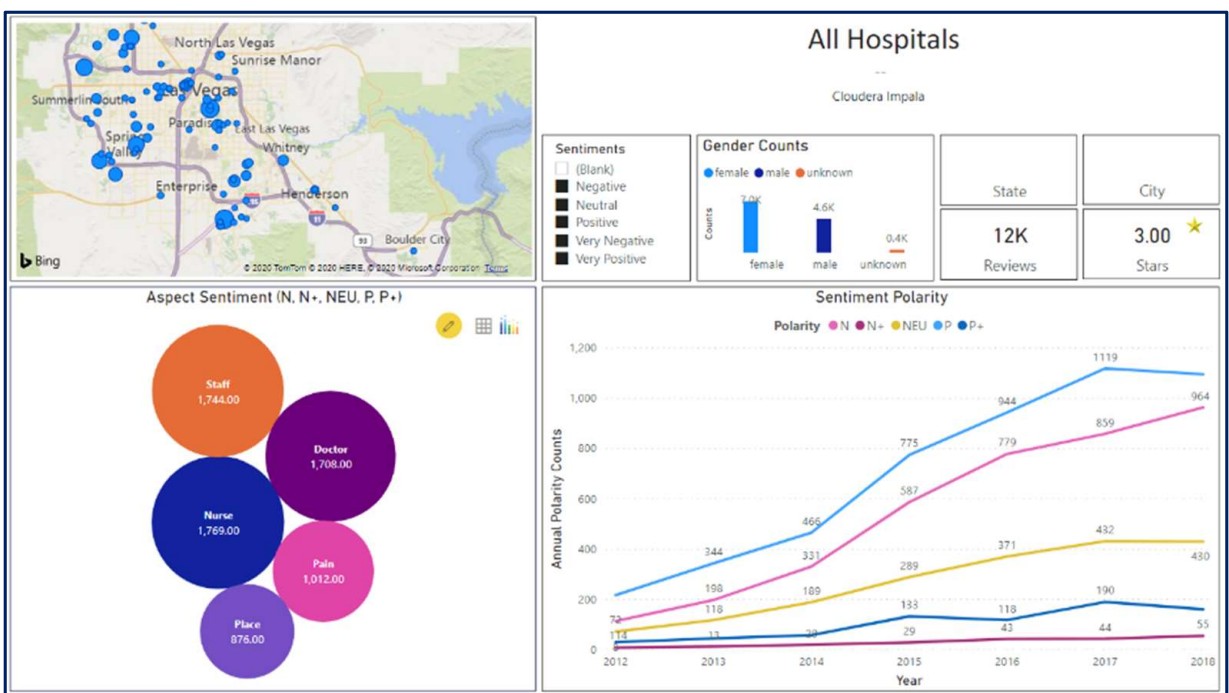

**Figure 10.** Main VoP dashboard.

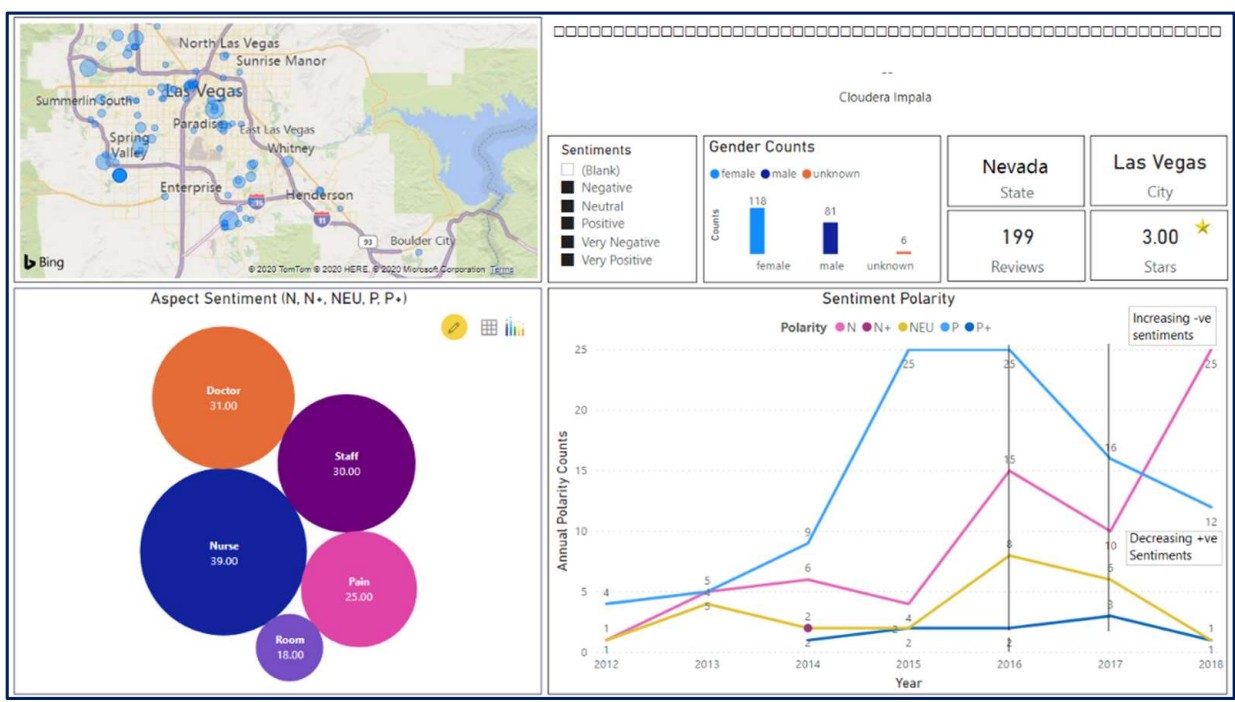

**Figure 11.** Analysis presentation (positive sentiment aspects).

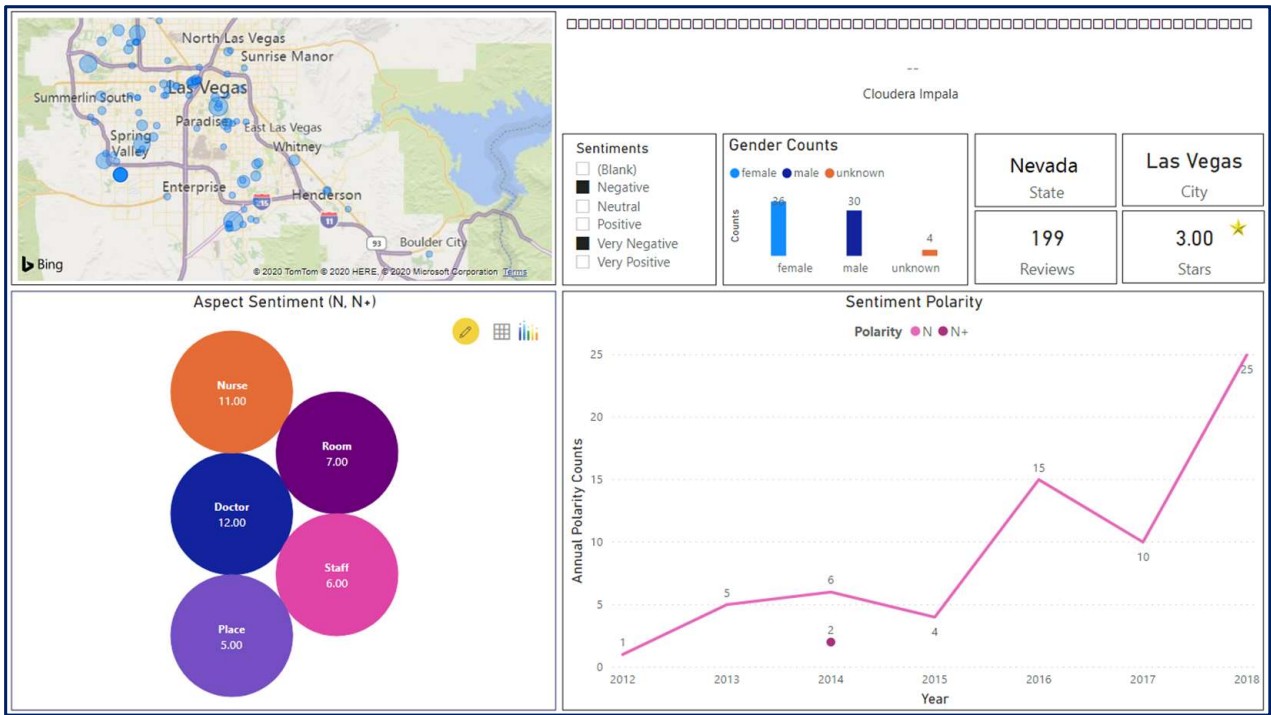

**Figure 12.** Negative sentiment aspects.

The bubble chart in Figure 11 shows that doctors, nurses, staff, and rooms are the dominant negative aspects of the case at hand. Similarly, Figure 13 shows the aspects of a positive sentiment case.

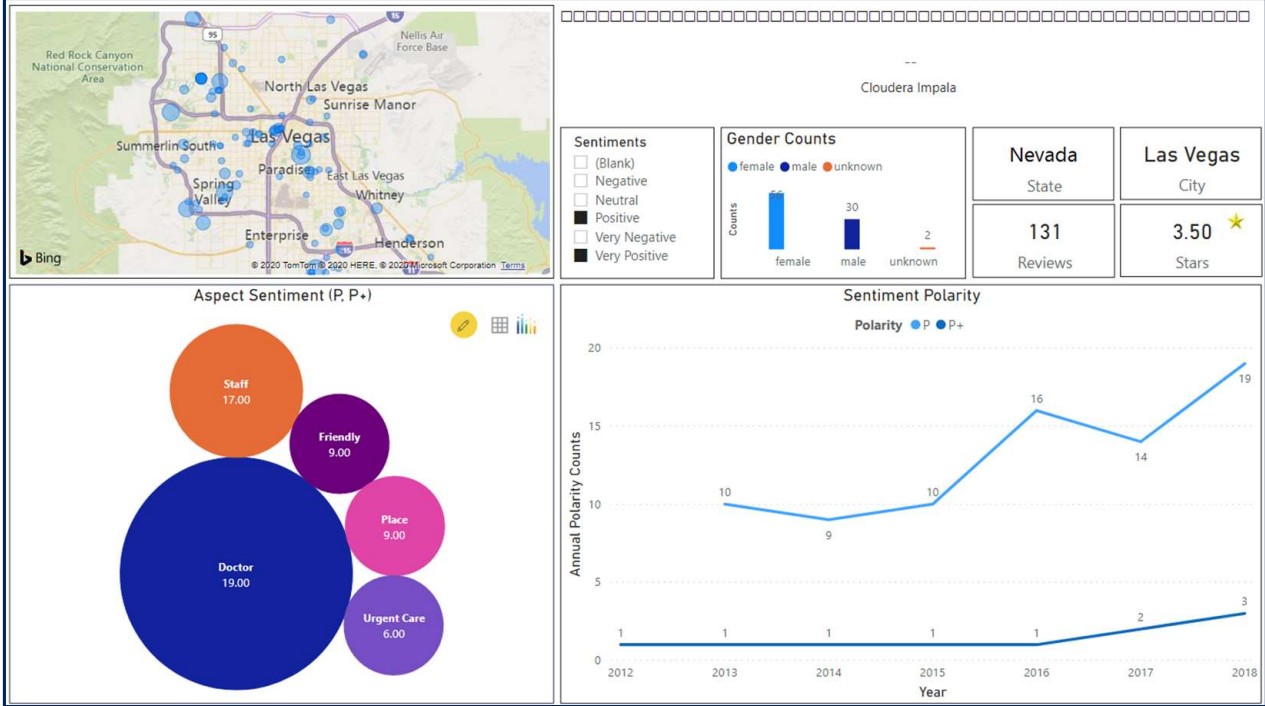

**Figure 13.** Positive sentiment aspects.

From the design science research perspective, SCDAP instantiation is an IS artifact [48], in which artifact evaluation is a critical step. To evaluate the potential contribution of SC-DAP in improving decision-making processes in the healthcare domain, we demonstrated SCDAP-VoP implementation to four senior managers and health technology professionals working in large healthcare organizations in Egypt. The demonstration is conducted through a PowerPoint presentation of the ideas and features provided by SCDAP-VoP. The presentation includes introductory slides about the fundamental ideas of BD, SCs, the proposed SCDAP framework, and VoP. The presentation's core part includes detailed information about using the main dashboard and how information supporting the decision-making process can be extracted from charts and graphs. At the end of the demonstration, the interviewees are asked to answer survey questions about how they find the SCDAP-VoP tool helpful in decision making, the challenges of implementing VoP in a real decision-making environment, and their opinions to enhance the outcomes of VoP.

The feedback of the interviewees is similar. In general, they agree in terms of the value of SCDAP-VoP as a framework for analyzing patients' reviews, their level of satisfaction, and their recognition of topics of interest and concern to them. Despite the consensus on the value of SCDAP-VoP, the interviewees made the following remarks about SCDAP-VoP, how to enhance the returned value, and the challenges in using it in the real environment:

1.  The integration of SCDAP-VoP analytics with other hospitals' IS will enable more detailed insights.

    -   Integration with a hospital's ERP will provide data that allow the clustering of patients according to their information (i.e., patient profiling).
    -   Integration with a hospital's healthcare system (Patient Health Record) will provide detailed indicative information about patients' responses to medication treatment.
    -   Integration with realtime health monitoring devices (e.g., smart bracelets) for elder patients and critical cases.

2.  It is recommended to combine unstructured data analytics (patients' reviews) with structured questionnaires (e.g., multiple choice questions or star ratings) for specific domains of satisfaction, such as empathy, service time, medical service value, and other standard key performance indicators, to enhance the outcome value of the analytics.

3.  Challenges might be faced during integration with different departments within hospitals and especially with patient records. Some might be concerned about breaching the Health Insurance Portability and Accountability Act (i.e., leakage of information to unauthorized parties).

4.  The distinction between model management and model aggregation is not straightforward. Model management is viewed as a more comprehensive function that comprises model aggregation.

5.  Model management and aggregation functionalities are valuable, as the tool provide comprehensive analytics these functionalities enable retention of extracted analytics. The interviewees expected that these functionalities' value would be more evident when studying patients' medical treatments' efficiency.

6.  The influence of rivals on social media (i.e., sponsored deliberate smearing campaigns) should be considered.

7.  More analytics about the reviewers themselves (demographics-based analytics) should be included.

8.  Encouraging patients to write their reviews in social networks or Web 2.0 applications in an objective manner is a challenging task that requires a non-trivial change of culture.

In response to the question "If SCDAP is implemented as a quality commercial software product, will you think to buy it?" two of the four interviewees responded, "I will think about it" and "I need more information." This is due to their expectation that there will be difficult finding the necessary technical personnel (a combination of data analysis expertise and the essential background in healthcare) to operate this system and use the available information. There is a common consensus about the potential benefits of incorporating BDA into the decision-making process.

## 8. Discussion

To discuss the value of SCDAP as a domain-independent decision support tool in light of the observations made by the interviewees, we will present the prototype implementation of SCDAP through the VoP use case from the following perspectives:

- Data acquisition
- End-user feedback analysis
- SCDAP design principles
- Generalizability
- Performance

### 8.1. Data Acquisition

SC data analysis is characterized by the necessity to connect the analysis with the data's spatial dimension. It is meaningless to analyze data without linking them to the spatial dimension. Of course, the level of spatial analysis changes according to business needs; it may require analysis at the level of the whole city, the level of districts, or even at the street level. The other natural dimension is the time dimension. Collectively, these two dimensions are called spatiotemporal coordinates.

Finding the appropriate datasets for SC analytics is one of the challenges in harnessing BDA in SCs [21,28]. Conducting analyses that include the spatial and temporal dimensions enables integrating many analyses from several domains. The spatiotemporal coordinates of the domains (d), services (s), events (e) and so on, which are meant to be tracked and analyzed, are therefore recorded with the datasets identified for analysis. This will support establishing a comprehensive reference spatiotemporal repository for the deduced analytical results (Figure 14). The granularity level of the spatiotemporal coordinates depends on the nature of the domain and the required analytics. Considering these characteristics in the data candidate for analysis in SCs, we added one more design principle to SCDAP design principles, which is the principle of spatiotemporal data.

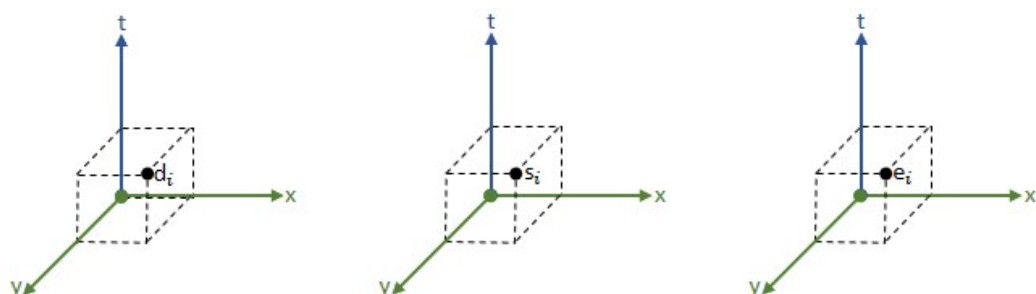

**Figure 14.** Spatiotemporal coordinates.

The value of the spatiotemporal dimensions appeared clearly in the VoP use case. Adding this dimension to the analysis enables decision-makers to focus on the place (analysis at the level of a specific hospital, group of hospitals, or even an entire state). The time dimension also enables the accumulation of historical analytics.

*8.2. End-User Feedback Analysis*

To analyze end-users' feedback listed in Section 7.4, we first project these comments into SCDAP design principles and BD 4Vs. The following table (Table 3) shows this projection of end user feedback into SCDAP design principles and BD 4Vs.

**Table 3.** Projection of end-user feedback into SCDAP design principles and BD 4Vs.

| Comment | SCDAP Design Principle | BD V Characteristics |
|---|---|---|
| a | DP2—Standardized data acquisition/access | Volume—Variety—Velocity |
| b | DP2—Standardized data acquisition/access DP3—Enabling both real-time and historical data analytics | Variety—Veracity |
| c | - | Veracity |
| d | DP5—Model management DP6—Model aggregation | - |
| e | DP5—Model management DP6—Model aggregation | - |
| f | - | Veracity |
| g | DP2—Standardized data acquisition/access | Veracity |
| h | - | - |

The previous table reveals the compatibility and harmony between end-users' desires in the specifications of the data analytics framework and the SCDAP design principle. This projection is valuable in that the interviewees only have a general background in BD and data analysis frameworks. In this regard, it is worth noting that SCDAP's first and fourth design principles (DP1—Principle of a layered design approach and DP3—Iterative and sequential processing) do not fall into the interest of end-users, as it is only the concern of system designers.

Interviewees gave considerable comments about the two functionalities related to model management and aggregation. First, the word "model" was confusing to them. Second, model aggregation is viewed as a sub-function of the model management functionality, and there is no need to treat it separately. Regarding the definition of these two functionalities in [10], we agree on these two observations, as the word "model" does not reflect the required meaning of the word properly; this word refers to the resultant data model or simply the resultant model. Therefore, it is more appropriate to rename the model management function as resultant model management. As for the second comment, model aggregation is meant to deal with (manage) various resultant data models. In that sense, we agree to include the aggregation functionality in the resultant model management. Accordingly, the two design principles (DP5—Principle of model management and DP6—Principle of model aggregation) will be merged into one principle: DP5—Principle of resultant model management. The SCDAP framework will also be modified to have only one functionality, resulting in model management (Figure 15).

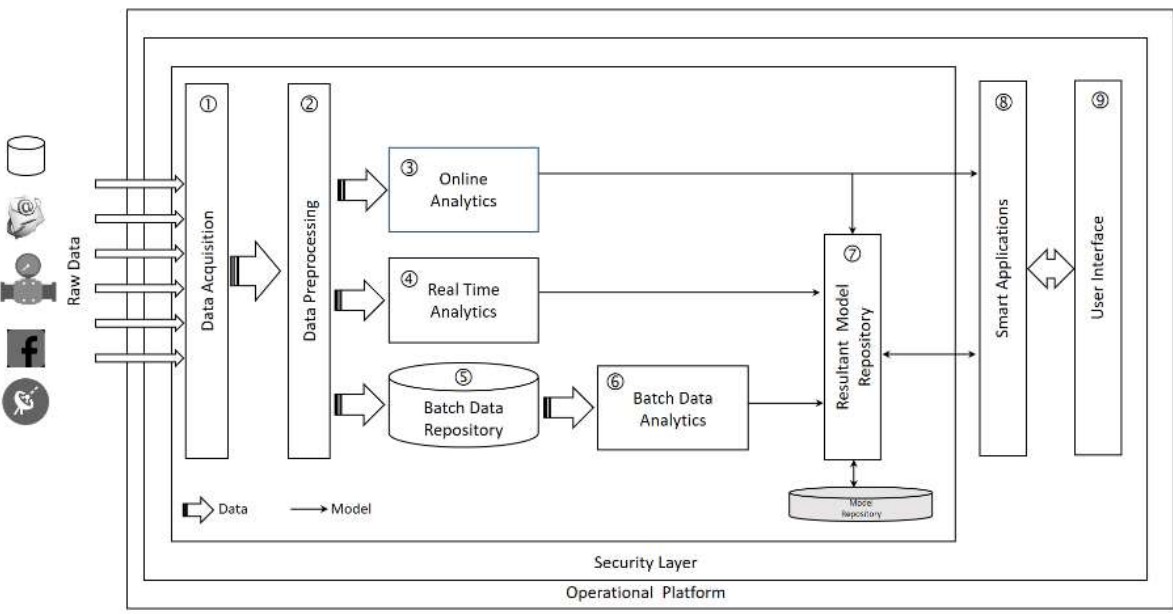

**Figure 15.** Modified SCDAP conceptual design.

### 8.3. SCDAP Design Principles

Based on the steps followed to instantiate the SCDAP prototype and the observations deduced from the VoP use case (Section 7), we reconsider SCDP design principles [10]. With the design experience of SCDAP prototype instantiation and its outcomes, the design principle of SCDAP falls into the category 3 design principle, according to [49]. The category 3 design principle in [49] is *action and materiality-oriented design principles*. This category's design principles prescribe what an artifact should enable users to do and how it should be built. Accordingly, we can reformulate the six design principles of SCDAP as follows:

The first four design principles will remain unchanged, whereas the fifth (DP5) and sixth (DP6) ones are merged into one principle as follows:

(DP5) Principle of resultant model management

This principle pertains to the system's ability to manage resultant data analysis models for future inquiries and analytics. This principle is supported by the ability to manage city reference data (e.g., maps).

Accordingly, the sixth principle will address data requirements:

(DP6) Principle of spatiotemporal data

This principle enables the establishment of a comprehensive reference repository for the resultant analytical models.

### 8.4. Generalizability

Generalization is a critical and vital attribute in designing an IS artifact [32,50]. In this regard, we refer to the third and fourth stages of the action design research (ADR) methodology proposed in [32]. The third stage of ADR, reflections and learning, refers to "*moving from building a solution for a particular instance to applying that learning to a broader class of problems.*" This is a vertical view of the class of problems that the artifact can address. This ideally applies to SCDAP implementation in VoP. As per the interviewees' feedback, SCDAP-VoP can be used to manage analytics about patients, services, and medical treatments, conditioned by the required data's availability.

The fourth stage of ADR, formalization of learning, refers to "*casting the problem-instance into a class of problems*." This is a horizontal view of the class of problems that the artifact can address. The principle this stage draws on is *generalized outcomes*, which suggests three levels of generalizations:

1. *Generalization of the problem instance*: This applies to SCDAP as a data-driven analytical framework designed to support decision-making in SCs. VoP is the special case, whereas patients are a subset of citizens, and data are citizen generated through social networks. Considering the inclusion of a spatiotemporal attribute to dataset candidates for analysis serves the purposes of SC analytics.
2. *Generalization of the solution instance*: The principal ingredients of the SCDAP instantiation prototype (data preprocessing, data analysis, the resultant model persistence) are general-purpose components designed to serve a broad range of problems.
3. *Derivation of design principles from the design research outcomes*: The reformulate design principles in Section 8.3 are based on the experience and knowledge gained through the design process if SCDAP and the implementation of VoP are generic principles (domain independent) and serve both scholars and practitioners.

*8.5. Performance*

As SCDAP realization and VoP use case implementation within this article's scope are just a prototype instantiation, we cannot claim that the measured performance figures could be used for benchmarking. However, storing and processing data in the Hadoop platform are generally a choice between HDFS, Apache HBase, or any other supported NoSQL database, such as Kudu from Cloudera. HDFS is appropriate for high-speed writes and scans, whereas HBase is ideal for random-access queries. Cloudera has announced that Kudu combines both HDFS and HBase's best features in a single package as general-purpose data storage for analytics and more. Of course, there are more alternatives for data storage on the Hadoop platform. The choice between these alternatives depends on the nature of the application. On the other hand, running queries and access to data can be performed through several tools, such as Hive or Impala. A comprehensive study to guide the choice between different technological alternatives is beyond the present article's scope.

**9. Conclusions**

The connection between SC and BDA has been investigated in academia. However, the proposition of domain-independent BDA frameworks that can serve a wide variety of SC stakeholders still needs further research. This article addresses the following research question: how can BDA be used as a data-driven decision-making enabler in SCs? To answer this question, we reviewed 26 articles concerning the domains they address in SCs. The findings show that only six articles (23%) proposed domain-independent BDA frameworks. One article proposed a BDA framework, SCDAP, which is domain independent and enables the preservation and interchange of extracted information between different stakeholders. A use case utilizing an evaluative prototype for SCDAP is instantiated to answer the research question. The prototype is implemented in a healthcare domain to analyze patients' reviews and identify their concerns and aspects of satisfaction (VoP). The findings revealed the resilience of SCDAP in managing the extracted models to preserve the inter-domain system interrelationship. As SCDAP is built using general-purpose software components, this resilience is extendable to SCs' smart domains. Another vital feature revealed by the findings is the necessity for a BDA framework to comprise a knowledge repository. This feature enables information exchange and extracted analytics between multiple stakeholders within the same domain or different domains.

The contribution of the research is multifold. From the academic perspective, it is a thoughtful preliminary attempt at bridging the gap in research related to domain-independent BDA frameworks instead of exclusive frameworks. Additionally, it introduces a novel concept of including a model repository in the BDA framework, opening new horizons in academic research in this field. This article presents a worthy idea for interchanging information between stakeholders in an SC from a practical perspective.

## 10. Limitations and Future Research Extensions

Although the findings on the use of BDA prototype instantiation and its implementation in an SC's VoP use case showed encouraging potential, the following limitations are worth considering in future implementations: combining both realtime and historical data analysis, integration of different general-purpose data analysis engines (e.g., Anaconda and TensorFlow), and benchmarking different data storage alternatives (e.g., HDFS, HBase, Kudu) to optimize performance, especially with realistic large volumes of data.

Future research, developing an impermeable end user accessibility scheme to prevent unauthorized access to the knowledge repository is a viable research direction. Developing standard messaging for exchanging models' metadata between different BDA frameworks is another research avenue. The latter will open the opportunity toward developing a standard messaging scheme for exchanging extracted models' metadata between various BDA frameworks.

**Author Contributions:** Conceptualization, A.M.S.O.; methodology, A.M.S.O.; software, A.M.S.O.; validation, A.M.S.O. and A.E.; formal analysis, A.M.S.O.; investigation, A.M.S.O.; resources, A.M.S.O.; data curation, A.M.S.O.; writing—original draft preparation, A.M.S.O.; writing—review and editing, A.M.S.O. and A.E.; visualization, A.M.S.O.; supervision, A.E.; project administration, A.M.S.O. All authors have read and agreed to the published version of the manuscript.

**Funding:** This research received no external funding.

**Acknowledgments:** The authors of this article would like to acknowledge RapidMiner and Meaning-Cloud for providing the necessary free academic licenses. The authors also recognize the contribution of the four interviewees and appreciate their time and valuable comments. Finally, the authors thank the development team, which contributed to the data preprocessing and coding of Python scripts.

**Conflicts of Interest:** The authors declare no conflict of interest.

## Appendix A. Analysis of Smart City Domains

| No. | Ref. | Domain Independent | Logistics | Mobility | Transportation | Energy | Price and Demand | Citizen Feedback | Data Security | Planning | Environment | Healthcare | Smart Building |
|------|-------|---|---|---|---|---|---|---|---|---|---|---|---|
| 1 | [7] | | | | | | | | | ✓ | | | |
| 2 | [8] | | | | | | | | | ✓ | | | |
| 3 | [10] | ✓ | | | | | | | | | | | |
| 4 | [11] | | | ✓ | | | | | | | | | |
| 5 | [12] | | | | | ✓ | | | | | | | |
| 6 | [13] | | | | | | | | | | ✓ | | |
| 7 | [14] | | | | ✓ | | | | | | | | |

| No. | Ref. | Domain Independent | Logistics | Mobility | Transportation | Energy | Price and Demand | Citizen Feedback | Data Security | Planning | Environment | Healthcare | Smart Building |
|---|---|---|---|---|---|---|---|---|---|---|---|---|---|
| 8 | [18] | ✓ | | | | | | | | | | | |
| 9 | [29] | | | | ✓ | | | | | | | | |
| 10 | [30] | ✓ | | | | | | | | | | | |
| 11 | [35] | ✓ | | | | | | | | | | | |
| 12 | [36] | ✓ | | | | | | | | | | | |
| 13 | [37] | ✓ | | | | | | | | | | | |
| 14 | [40] | | | | | | | ✓ | | | | | |
| 15 | [51] | | ✓ | | | | | | | | | | |
| 16 | [52] | | | ✓ | | | | | | | | | |
| 17 | [53] | | | ✓ | | | | | | | | | |
| 18 | [54] | | | | ✓ | | | | | | | | |
| 19 | [55] | | | | | ✓ | | | | | | | |
| 20 | [56] | | | | | | ✓ | | | | | | |
| 21 | [57] | | | | | | | | ✓ | | | | |
| 22 | [58] | | ✓ | | | | | | | | | | |
| 23 | [59] | | | | | | | | | | | ✓ | |
| 24 | [60] | | | | | ✓ | | | | | | | |
| 25 | [61] | | | | | | | | | | | | ✓ |
| 26 | [62] | | | | | | | | | | ✓ | | |
| | Total | 6 | 2 | 3 | 3 | 3 | 1 | 1 | 1 | 2 | 2 | 1 | 1 |

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
