# Peer review of "Smart Cities and Big Data Analytics: A Data-Driven Decision-Making Use Case"

_smartcities, doi:10.3390/smartcities4010018_

Round 1

Reviewer 1 Report

  1. English should improve. This paper should proofread by the professional proof reader.
  2. Authors of this manuscript are used two terms for big data, sometimes they used BD and sometimes Big Data. 
  3. Research method is weak.
  4. Results are not satisfactory.
  5. Add some more latest references.

Reviewer 2 Report

The paper presents results on the intersection of big data science and smart cities what no doubt is a hot topic.

The main concern though with the paper presented is that at the very begining it shows as a review paper but later it seems rather an evaluation of a determined tool found in the review process.

Kind of the contribution of the paper are confusing as a consequence of a not clear goal of the research.

Regarding the use case presented if the authors want to extract conclusions then comparison with other frameworks are missing.

On the other hand the use case chosen does not show smart cities data and complexity and rather seems to be an analysis of data gathered of social media.

In relation to the review part and analysis of the state of the art it also looks like the keywords used for searching are too few and more terms should be used to really reflect research on big data and smart cities.

In relation to the use case some down-level coding details are not really required and more details are missed regarding performance, comparison, ...

The lack of a clear objective together with  a lack of scientific rigor makes the contribution doubtful.

Reviewer 3 Report

The topic of the article is interesting and it is possible to distinguish that the contribution exists, but the structure does not correspond to the structure of a scientific article in this and other scientific journals. The article needs to be completely restructured to be clear and readable. The journal template is available on the site and needs to be applied to your article. Too many chapters and subchapters is unnecessary and leaves quite a lot of questions when reading the paper. The basic chapters are:

1. Introduction - in which it is necessary to describe the topic of your research, state the hypothesis and the basic objectives of the work. Equally, it can be divided into subchapters 1. Aim of the research and 2. Literature review (Review of the current scientific literature). In the review of the previous scientific literature, it is necessary to explain in detail what has been done in the field of research so far, and in a few sentences to state what has been proposed by your research.

2. Methods and methodology - this is a central part of your scientific contribution. It is possible to cite the existing methodology, but the essence should be put on what you suggest. Here you need to clearly and in detail specify your research on all key points.

3. Results - in this section you validate your proposed methodology and conclude whether the hypothesis is confirmed. The results need to be described both textually and graphically.

4. Discussion - this chapter provides answers to questions about how the proposed methodology has improved existing scientific facts. In this part, it is again necessary to refer to the papers listed in the literature review and state whether your work provides a quality scientific step forward in relation to them.

5. Conclusion - in this section it is necessary to comment on the results, again refer to the hypotheses and supporting objectives. It is necessary to comment on the results and give guidelines for future research.

Figures 4, 8, 9, 10, 11, 12 and 14 are not visible. Tables 1, 2 and 3 are not made according to the template.

In the very beginning, it is necessary to satisfy the basic principles of writing a scientific paper so that it can be reviewed. This way it looks more like a seminar paper than a serious scientific article.

Round 2

Reviewer 1 Report

Now, the paper is in the acceptable form.